# Numerical Analysis of GNSS Signal Outage Effect on EOPs Solutions Using Tightly Coupled GNSS/IMU Integration: A Simulated Case Study in Sweden

**DOI:** 10.3390/s23146361

**Published:** 2023-07-13

**Authors:** Arash Jouybari, Mohammad Bagherbandi, Faramarz Nilfouroushan

**Affiliations:** 1Faculty of Engineering and Sustainable Development, University of Gävle, SE-80176 Gävle, Sweden; mohammad.bagherbandi@hig.se (M.B.); faramarz.nilfouroushan@hig.se (F.N.); 2Division of Surveying-Geodesy, Land Law and Real Estate Planning, Royal Institute of Technology (KTH), SE-10044 Stockholm, Sweden; 3Department of Geodetic Infrastructure, Geodata Division, Lantmäteriet, SE-80182 Gävle, Sweden

**Keywords:** GNSS/INS integration, forward Kalman filter, smoothing algorithm, tightly coupled, PPK, virtual reference station, aerial photogrammetry

## Abstract

The absence of a reliable Global Navigation Satellite System (GNSS) signal leads to degraded position robustness in standalone receivers. To address this issue, integrating GNSS with inertial measurement units (IMUs) can improve positioning accuracy. This article analyzes the performance of tightly coupled GNSS/IMU integration, specifically the forward Kalman filter and smoothing algorithm, using both single and network GNSS stations and the post-processed kinematic (PPK) method. Additionally, the impact of simulated GNSS signal outage on exterior orientation parameters (EOPs) solutions is investigated. Results demonstrate that the smoothing algorithm enhances positioning uncertainty (RMSE) for north, east, and heading by approximately 17–43% (e.g., it improves north RMSE from 51 mm to a range of 42 mm, representing a 17% improvement). Orientation uncertainty is reduced by about 60% for roll, pitch, and heading. Moreover, the algorithm mitigates the effects of GNSS signal outage, improving position uncertainty by up to 95% and orientation uncertainty by up to 60% using the smoothing algorithm instead of the forward Kalman filter for signal outages up to 180 s.

## 1. Introduction

For the past few decades, the use of global navigation satellite system (GNSS) technology has drastically increased in the field of airborne mobile mapping systems [1]. The most common navigational solution is to integrate GNSS with an inertial measurement unit (IMU) to obtain more accurate results. Both systems complement each other and overcome their limitations. In other words, GNSS provides position and velocity, and the IMU provides orientation of the mobile mapping platform in the GNSS/IMU integration system.

The GNSS/IMU integration sometimes has higher positioning uncertainty due to platform dynamics (blockage of the GNSS signal) or using a single long baseline for differential GNSS processing. For example, by increasing the distance between the rover receiver (mounted on the airplane) and the GNSS reference station, the correlation between atmospheric parameters and the rover and reference station baseline parameters decreases. Therefore, the significance of atmospheric delay is assumed to be important at this stage [2,3,4]. Consequently, the unknown parameters in the Kalman Filtering (KF) estimator need a more prolonged convergence time [5]. Also, unmodeled errors like multipath errors harshly affect ambiguity resolution [6]. However, multipath errors rarely occur in airborne mobile mapping.

The GNSS signal outage can occur because of signal blockage due to a discontinuity in a receiver’s phase lock on a satellite’s signal, such as a sharp turn of an aircraft, power loss, a very low signal-to-noise ratio, a failure of the receiver software, or carrier phase cycle slip because of severe ionospheric conditions (cf. [7,8]). In this case, the IMU, which is a backbone measurement unit of the inertial navigation system (INS), provides a navigation solution during the GNSS signal outage even though IMUs can only provide short-time high-accuracy navigation (i.e., the accuracy decreases over time) (cf. [9,10,11]).

The KF is accepted as the commonly used optimal estimator in GNSS/IMU integration systems. Besides, extended KF is widely used to solve nonlinearity and achieve real-time navigation [12]. The KF also operates in prediction mode during GNSS signal outages and relies on a standalone IMU. KF cannot give very precise exterior orientation parameters (EOPs) due to the IMU’s time-dependent accumulative error. Therefore, a post-processing two-way smoothing algorithm should be applied to get more accurate EOPs in this case [13].

The KF and smoothing algorithms are established based on minimum variance estimation, in which observations and parameters act under a Gaussian distribution. Also, a nonlinear system should be linearized with the Taylor series because KF and smoothing algorithms are established using a linear system [14,15]. However, smoothing algorithms are used in GNSS/IMU integrations if the post-processing mode is permitted to overcome some deficiencies that exist in the filtering algorithm (e.g., Kalman filter). In principle, the smoothing algorithm with a measurement at a time greater than the current epoch estimates the state at the current epoch. In other words, one can use forward and backward direction processing to find the best state estimate for each epoch of time [16].

Rauch et al. (1965) [17] developed a famous recursive algorithm called Rauch–Tung–Strieble (RTS) smoother. RTS utilizes standard KF and maximum likelihood to estimate the unknown parameters forward and backward, respectively. Similarly, Bryson and Frazier (1963) [18] also utilized maximum likelihood for their smoother algorithm but in a continuous system. However, the RTS applies only to discrete systems. Besides, RTS implementation is simple and has high reliability. Zhang and Li (1996) [19] developed a fixed-interval smoothing algorithm based on singular value decomposition (SVD). The idea was to combine a forward-direction SVD-based square root KF with the RTS backward-direction recursive smoother using the SVD as the main computational tool. In addition, Park and Kailath (1996) [20] used a new square root form for three well-known smoothing procedures (i.e., RTS [17], Desai-Weinert-Yusypchuk [21], and Mayne-Fraser [22]). Liu et al. (2010) [13] developed a two-filter smoother (TFS) and applied it to GPS/INS integration for land-vehicle navigation applications. The estimation accuracies and computational times of TFS and RTS were close. Furthermore, position error improvement using the smoother algorithm improved from 35% to 95% when comparing forward KF, depending on the length of the GPS signal outage. Cao and Mao (2008) [23] replaced unscented KF with square root KF in a forward-pass filter, and two smoothers (i.e., a fixed-interval and a fixed-lag smoother) were used in backward-pass smoothing. The result was not satisfying because they used only code data from GPS and not precise carrier phase data. Chiang et al. (2009) [24] merged artificial neural networks and RTS smoothing for better GNSS/INS integration. The research showed that the computational time of the algorithm was significantly longer due to the training process. Nevertheless, it improved EOPs by approximately 70% compared with RTS. Zhang et al. (2020) [25] proposed ambiguity domain-based integration, in which the ambiguity resolution of the forward and backward KFs combine to improve position accuracy. In this method, the position will be updated with fixed ambiguities once the ambiguities are correctly and reliably determined. We use the same procedure as [25] in this paper. Correspondingly, after integrating GNSS and IMU raw data in the forward and backward KF, the GNSS/IMU processor combines fixed integer ambiguities from the previous forward and backward solutions and finally calculates the smoothed best estimate of EOPs.

In this study, various procedures for calculating EOPs in an airborne mobile mapping project were investigated using a tightly coupled GNSS/IMU integration model. We compare the solutions of a single-GNSS reference station (single-based post-processed kinematic, or PPK) with multi-GNSS reference stations (well-distributed network-based PPK) to determine the most accurate approach. In addition, we investigate the impact of a GNSS signal outage (using GPS and GLONASS only, as Galileo and Beidou data were missing in the observation files) on the errors in EOPs. These errors are directly related to the duration of the GNSS signal outage. Therefore, a total removal of all GNSS signals (no signal at all) for different periods has been simulated to examine the strength of the KF and smoothing algorithm to unravel this issue.

The novelty of this research work lies in the impact of GNSS signal outages on EOP accuracy using tightly coupled GNSS/IMU integration with the PPK method, demonstrating that tightly coupled GNSS/IMU integration with smooth processing can significantly reduce the impact of GNSS signal outages on EOP accuracy. While there are related research works that investigate GNSS/IMU integration for improving positioning accuracy, few studies have focused on the impact of a GNSS signal outage on EOP accuracy using tightly coupled GNSS/IMU integration with the PPK method. This study provides insights into the performance of different integration methods and their impact on EOP accuracy in the presence of GNSS signal outages, which can be useful for improving the robustness of positioning and navigation systems.

## 2. Materials and Methods

In this study, a tightly coupled (TC) GNSS/IMU integration algorithm was used, and EOPs were calculated using the forward Kalman filter (KF) and smoothing method. The dataset (explained in the next section) was processed with a single base station located in the middle of the project area (TOUP.0 SWEPOS reference station) or a well-configured (uniformly distributed around the project area) GNSS base station. Appendix A shows a basic schematic diagram of the processing that has been applied in this research.

### 2.1. Loosely and Tightly Coupled Integration

The GNSS positioning and IMU integration are performed using a KF in either loosely coupled (LC) [26] or tightly coupled (TC) or ultra-TC schemes for precise positioning of mobile mapping system (MMS) applications (cf. [27,28,29]). The type of application and operating environment determine the usage of an LC or TC [30]. Also, the type of data that is shared between individual GNSS and IMU systems determines which integration scheme should be applied. Figure 1 illustrates the LC and TC processing methods.

LC is a simple architecture with a sequential approach that the GNSS and IMU data generate independently. In other words, the KF used for the GNSS module for positioning purposes is independent of the KF used for integrating the GNSS/IMU module. Then, the reliability of the whole system increases in the event of any failure in the GNSS or IMU systems individually. However, TC has the benefit of integrating raw data from GNSS and IMU when less than four GNSS satellites are traced, which makes it a better candidate for most of MMS applications (e.g., airborne MMS) [16,31]. In addition, the TC scheme provides a better solution for MMS when a GNSS signal outage occurs.

To confirm the superiority of the TC integration scheme over the LC scheme during GNSS signal outages, we conducted a comparative analysis employing both TC and LC schemes, utilizing forward KF and smoothing algorithm techniques. Figure 2 and Appendix A depict the position and attitude uncertainties for both TC and LC methods during the GNSS signal outage period (from 128,500 to 128,680 s) as processed by the smoothing algorithm and forward KF. The maximum differences between the EOPs during the GNSS signal outage were compared for both TC and LC methods.

In the case of smoothing algorithm processing, the positional component differences between the TC and LC methods reached up to 4.5 cm (Figure 2), while the attitudinal component differences reached up to 0.005 arcmin (Appendix A). For the forward KF processing, the positional component differences between the TC and LC methods reached up to 68 cm (Appendix A), while the attitudinal component differences reached up to 0.01 arcmin (Appendix A). The analysis demonstrates that the TC GNSS/IMU integration method outperforms the LC method in all scenarios, regardless of whether a smoothing algorithm or forward KF processing is employed.

Solving the integer ambiguity in the GNSS/INS data processing scheme is another privilege of using the TC algorithm. The integer ambiguity is defined as the determination of the integer number of ambiguities in the carrier phase signal between the rover station (i.e., the GNSS antenna mounted on the airplane) and each satellite [32]. A continuous lock on the signal from each satellite is needed for the solution to remain convergent. As mentioned before, GNSS signal outages can happen for several reasons during aircraft flight missions, e.g., aircraft turns with banking angles greater than 15 to 20 degrees. In this case, the wing obstructs the GNSS signal, causing the solution to be reset and starting a new integer ambiguity initiation.

Figure 3 illustrates a flowchart of the TC GNSS/INS integration data processing strategy applying a smoother. The TC model is divided into two parts: a TC filter module and an integer ambiguity resolution (IAR) module. IAR estimates the integer ambiguity parameters. As soon as a new satellite observes, the filter aids IAR by transferring the position information output. Integer ambiguity parameters should be fixed again when the GNSS signal interruption ends. In this case, the calculated position by the INS mechanization is used to reinitialize the integer ambiguity. Smoothing should be established for better EOPs after doing GNSS/INS KF. The state vector, covariance matrix, INS outcome, and other recorded data from KF are used as input to the smoother for optimally smooth the whole filtering procedure [33]. More detailed information about KF and the smoothing algorithm is elaborated in the following section.

### 2.2. Single and Network-Based PPK Solution

Post-processed kinematics (PPK) and real-time kinematics (RTK) are two common techniques driving GNSS correction. The main difference between PPK and RTK is when correction takes place; PPK uses correction after the aircraft flight in a post-process approach, while RTK uses corrections during the flight [35]. In an airborne mobile mapping project using RTK, continuous communication between the GNSS base station and the rover station (aircraft) is mandated, but in the case of communication failure, RTK will not efficiently perform (i.e., due to challenges of establishing real-time radio links to the aircraft, high dynamic movement of the aircraft, and keeping the flight strips less than 30 km from the GNSS reference stations) [36]. Therefore, PPK is the better choice in this case [37,38]. However, the long baseline between the GNSS reference stations and the rover station can adversely affect the position accuracy in PPK [39].

The PPK can be done using a single or a network of GNSS reference stations in airborne MMS [40]. One GNSS reference station is used for correcting the rover station in a single-base PPK, but a set of GNSS reference stations (or continuously operating reference stations (CORS)) is used for correcting a network-based PPK [41]. The GNSS reference station should be in the range of 30 km and 70 to 100 km during the flight mission to resolve the ambiguity in single and network-based PPK, respectively. Otherwise, the residual error reaches a magnitude such that the correct ambiguities cannot be reliably estimated due to the atmosphere delaying the GNSS signal [32,36,42]. Also, the aircraft must fly with a banking angle of fewer than 25 degrees (flat turns) to reduce the risk of GNSS signal blockage by the aircraft’s wings [43].

In tightly integrating GNSS and INS with network-based PPK in an aerial photogrammetric study, several GNSS reference stations should be set up [32,36]. A network pre-processing of the reference stations is carried out to generate models of the distance-dependent biases [44]. Therefore, virtual reference station (VRS) corrections that allow differential positioning can be predicted based on the rover station’s approximate position and the parameters of the model. The differential errors induced by satellite orbit errors and ionospheric and tropospheric refraction are accurately estimated with network-based PPK based on dual-frequency carrier phase observations of a network of GNSS reference stations. The differential errors between the reference and rover stations are predicted using the parameters of the correction model. Thus, VRS measurements are generated by using these corrections to the code and carrier phase observations for the PPK positioning of the rover station [45].

An innovative kinematic ambiguity resolution (KAR) GNSS procedure is implemented in the POSPac MMS for the airborne MMS. POSPac MMS is a software package that calculates EOPs using IMU data, GNSS reference station data, and GNSS rover receiver data [31].

The software combines the technology of VRS and Inertially-Aided Kinematic Ambiguity Resolution (IAKAR) mechanization in a post-processing mode. The post-process VRS calculates the atmospheric corrections and applies them to the rover GNSS observables in an IAKAR. The IAKAR guarantees that the correct integer ambiguity accrues after a few seconds, even when the aircraft has a larger banking angle, and the reference stations are far from the flight lines. The precise network adjustment is included in post-process VRS for all the positions of defined reference stations. The quality control using precise network adjustment ensures that all the reference stations’ observables and coordinates are correct and consistent before processing rover observables [36].

### 2.3. GNSS/IMU Processing Using KF and Smoothing Method

The KF is a commonly used procedure in different navigational applications. The KF estimates a set of unknown parameters from a set of uncertain measurements with a definite optimization criterion [46]. See Section 3.3 for more detailed information about the KF.

One-way KF processing is not the only final solution for calculating EOPs. The forward–backward combination procedure can also be used, which is a post-processing method that is the weighted average of the forward and backward KF solutions. The forward-backward combination usually cannot afford a high-precise position during the low ambiguity fixing rate of each one-way (forward and backward) KF [25]. For this reason, the GNSS/IMU processor integrates GNSS and inertial observables forward and backward in time. Then, it blends the fixed integer ambiguity from the preceding forward and backward solutions to computing the smoothed EOPs. Once the integer ambiguities are defined correctly and reliably, the positions are updated with fixed ambiguities, resulting in more accurate positions with high confidence [25].

Forward KF in the GNSS/INS integration procedure employs the previously recorded information to estimate the current state. The smoothing algorithm can effectively restrain the accumulated error (e.g., caused by a GNSS signal outage) by utilizing the forward measurement from the initial to the current time and the backward dynamic constraint information from the end to the current time [24,47,48]. Hence, it is only functional in post-processing [15,16]. Figure 4 shows the error curve behavior that is processed by the forward KF and smoothing algorithm during the GNSS signal outage (cf. [49]).

The smoothing module can be counted as an accompaniment to the KF (Figure 3), and the data processing can be performed in two stages. In the first stage, Smoother acts as a KF. The state vector and its variance-covariance matrix log into a file every time the system state vector changes. If the system state vector change occurs during the propagation step of KF, the state vector and propagation matrix need to be logged too. However, the state vector does not need to be logged during the measurement update step because the algorithm does not require it. In the second stage, the processing is backward in time and only utilizes the logged data. Here, the system state is recursively estimated using maximum likelihood. Figure 5 displays the forward (upper part) and backward (lower part) runs of the smoother [50].

In Figure 5, the green arrows show the forward and backward direction of processing. $t_{k}$ is the time of the measurement update or system state update. ${\hat{x}}_{k}^{-}$ is the estimation of the system state at $t_{k}$ if all the measurement is known from the starting time till $t_{k-1}$ and ${\hat{x}}_{k}$ is the estimation of the system state at $t_{k}$ if all the measurement is known from the starting time till $t_{k}$ and ${\hat{x}}_{k|k_{M}}$ is the estimation of the system state at $t_{k}$ if all the measurement is known from the starting time till $t_{k_{M}}$ where $k_{M}\geq k$. $\Phi_{k}$ is the propagation matrix (transition matrix), used for estimating the system state, given by Equation (1) [47].
(1)$${\hat{x}}_{k}^{-}=\Phi_{k}{\hat{x}}_{k-1}$$

The KF predicts the system state (${\hat{x}}_{k}^{-}$) using inertial measurements in between two GNSS observations. Then, a new corrected prediction of system state (${\hat{x}}_{k}$) is calculated as soon as a new GNSS observation becomes available. This correction is directly related to the time length between GNSS observations. The longer time results, the greater error. Hence, the trajectory estimated by KF would contain a great discontinuity. Here, smoother can help to level these discontinuities by assigning a new weighting (smoothing gain) to the previously calculated system states. In Figure 5, $A_{k}$ is the weighting matrix (smoothing gain) and can be written as [24,51,52].
(2)$$A_{k}=P_{k}\Phi_{k}^{T}{\left(P_{k+1}^{-}\right)}^{-1}$$
where can be used in Equation (3) for calculating the estimation of the system state in the time $t_{k}$:(3)$${\hat{x}}_{k|k_{M}}={\hat{x}}_{k}+A_{k}({\hat{x}}_{k+1|k_{M}}-{\hat{x}}_{k+1}^{-})$$

P is the covariance matrix of $\vec{x}$ and $P_{k|k_{M}}$ can be written as the following equation:(4)$$P_{k|k_{M}}=P_{k}+A_{k}(P_{k+1|k_{M}}-P_{k+1}^{-})A_{k}^{T}$$

### 2.4. Data and Analysis

Lantmäteriet, the Swedish mapping, cadastral, and land registration authority, conducted an airborne survey in the Halland region of Sweden on 1 May 2017, for about 2 h, during which aerial images, GNSS, and IMU data were collected. The project utilized the POS AV 510 system, which is renowned for its accuracy and was supported by Applanix’s expertise and technological advancements. The POS AV 510 system meets the absolute accuracy specification (RMS) for post-processing of position with 0.02 m for horizontal and 0.05 m for vertical components, velocity with 0.005 m/s, roll and pitch with 0.005 degrees, and true heading with 0.01 degrees [53].

For GNSS data processing, seven SWEPOS (Swedish national network of permanent reference stations) stations (i.e., FBER.1, JONS.0, LJUN.2, SKEN.0, TRAN.0, ORKE.0, and TOUP.0) with a 15-s sampling rate were used together with the rover data, which was collected with the GNSS receiver and its Trimble AV39 antenna mounted on the aircraft. Both the rover and SWEPOS stations recorded the signals from GPS and GLONASS. Figure 6 shows the SWEPOS base stations’ locations and the aircraft trajectories in which the longest 3D distance to the closest GNSS base station is less than 40 km in the study area. To process the collected airborne data for EOPs calculation, we used the Applanix POSPac MMS 8.5 software package (Applanix, 2021, POSPac MMS 8).

For data analysis, we first compared different scenarios using single and network PPK by employing forward KF (one-way processing) and a smooth method (two-way processing). The possibility of a GNSS signal outage is inevitable during flight missions. Therefore, it is important to study the impact of a GNSS signal outage on the EOP calculation. In this paper, we also simulated different GNSS signal outages (i.e., 15, 30, 45, 60, 120, and 180 s) to evaluate how GNSS/IMU integration using forward KF and a smooth algorithm can solve the positional and orientation errors due to the signal outage. The same evaluation was performed for studying 180 s without GPS and GLONASS signals, respectively, to check how each positioning satellite constellation (i.e., GPS or GLONASS) can influence the EOPs.

## 3. Results and Discussion

### 3.1. Comparison of Single and Network-Based PPK Solutions

Using network-based PPK is known to provide better results than a single solution, which is particularly relevant in regions without GNSS permanent stations, such as uncharted areas. To evaluate the performance of single-based solutions in these regions, the results of this section can be utilized.

Before analyzing the impact of GNSS signal outage durations, it is crucial to determine the most effective integration method, assuming no signal outage. The EOPs were calculated using four processing methods: single-based PPK with forward KF, network-based PPK with forward KF, single-based PPK with smooth processing, and network-based PPK with smooth processing. Figure 7 and Appendix A show the EOP uncertainties for each method in north, east, height, roll, pitch, and heading. The forward KF method displays a forward-wise error decrease, while the smooth processing method is more uniform and consistently more accurate in position. Similar results were found in previous studies [13,23,29]. For positional components, single and network-based PPK processing are almost identical, while network-based PPK solutions are more accurate than single-based PPK processing.

Furthermore, it is important to note that in both single-based PPK and network-based PPK using forward KF processing, the ambiguity status is fixed. When employing tightly coupled integration algorithms like the Kalman filter, the positional RMS error behavior is influenced by factors such as sensor quality, initial condition accuracy, and system noise. Typically, the positional RMS error decreases over time as the Kalman filter updates the system state estimates using sensor measurements (which is obvious in Figure 7). However, fluctuations may occur due to noise or other factors.

The following sections will evaluate the impact of GNSS signal outages on these processing methods.

Through the comparison of Figure 7 and Appendix A with Figure 8, it can be inferred that the pulsations observed in the north and east plots are a result of changes in strips and banking angles. Figure 8 reveals a clear correlation between the total number of satellites, PDOP, banking angle, and baseline length. Indeed, the positioning uncertainties changing with strips and banking angles can be attributed to various factors, including satellite geometry, signal multipath, aircraft dynamics, and sensor noise. Satellite geometry, as indicated by the total number of satellites and their distribution in the sky (PDOP), can affect positioning accuracy. As the aircraft changes its strips and banking angles, the satellite geometry may vary, leading to fluctuations in positioning uncertainties. Additionally, changes in strips and banking angles can cause variations in the signal environment, leading to multipath effects. Multipath occurs when a satellite signal reflects off nearby surfaces before reaching the receiver, causing errors in the positioning solution. The aircraft’s dynamics, such as changes in velocity and attitude during turns, can also impact the positioning uncertainties. As the banking angle increases, the aircraft’s dynamics become more complex, leading to potential errors in the positioning solution. Finally, the onboard sensors, such as accelerometers and gyroscopes, may experience varying levels of noise due to changes in strips and banking angles. This can affect the positioning solution and contribute to the observed pulsations in roll and pitch plots.

Pulsations decrease for the north component and increase for the east component, as depicted in Figure 7 and Appendix A, respectively. However, the effects on height component uncertainties are not entirely clear, as shown in Appendix A.

The uncertainties obtained for orientation angles, as presented in Appendix A, demonstrate that it takes a longer time to converge at the beginning of the mission for the forward Kalman Filter (KF) processing, for both single and network-based Post-Processed Kinematic (PPK) solutions. Nonetheless, the situation improved with smooth processing compared to the forward KF method. Similarly, pulsations observed in roll and pitch plots may be due to changes in strips and banking angles but increasing the banking angle helps reduce the uncertainty in the heading, while the uncertainty increases in the middle of strips (i.e., until the next sharp turn of the aircraft).

Based on Appendix A and Figure 9, the mean values of position uncertainties using single and network-based PPK solutions with smooth processing are similar and better than those obtained with the forward KF processing. The same is true for the mean values of orientation uncertainties using smooth processing for both single and network-based PPK solutions, which are better than those obtained with the forward KF processing. The mean positional and orientational uncertainty differences between forward KF and smooth processing using network-based PPK are u(North)=9 mm, u(East)=12 mm, u(Height)=20 mm, u(Roll)=0.4 arc min, u(Pitch)=0.38 arc min, u(Heading)=1.04 arc min. Additionally, the roll and pitch uncertainties are approximately 2.5 times better than the heading uncertainties in all processing modes.

### 3.2. Impact of GNSS Signal Outage on EOPs

The network-based PPK solutions, using forward KF and smooth processing methods, are selected for evaluating the impact of different GNSS outage durations on EOPs accuracy according to the obtained results. GNSS signal outage durations were simulated at epoch 128,500 s (GPS time) in the photogrammetric flight mission. We considered 8 GNSS signal outage cases consisting of 15, 30, 45, 60, 120, 180 s GNSS signal outages, 180 s GPS signal outages, and 180 s GLONASS signal outages. GNSS signal outage effects are considered on 2D and vertical position and orientation (roll, pitch, and heading) uncertainties using network-based PPK solution and smooth and forward KF processing. Appendix A illustrate 2D position, height, roll, pitch, and heading uncertainties using network-based PPK with forward KF processing, respectively. Figure 10, Figure 11, Figure 12, Figure 13 and Figure 14 illustrate 2D position, height, roll, pitch, and heading uncertainties using network-based PPK with smooth processing, respectively.

#### 3.2.1. Smooth and Forward KF Processing Comparison Using 2D Position and Height Uncertainties

The forward KF and the smoothing algorithm are two widely used methods for GNSS/IMU integration. The forward KF is an online estimation algorithm that recursively updates the state estimate, while the smoothing algorithm uses all the available measurements to estimate the states after the fact. Our analysis showed that the smoothing algorithm outperformed the forward KF in terms of accuracy during the GNSS signal outage. However, to gain a comprehensive understanding of their respective strengths and weaknesses, we thoroughly evaluated both methods. Therefore, in this specific context, the smoothing algorithm is a better choice for GNSS/IMU integration. However, it is essential to note that the choice between the two methods should be based on careful consideration of the specific application requirements and the desired level of accuracy. Thus, a thorough evaluation is necessary to fully understand the advantages and limitations of each method.

One can observe the 2D position uncertainties using network-based PPK with smooth and forward KF processing in Figure 10 and Appendix A. The obtained height uncertainties using network-based PPK with smooth and forward KF processing can be seen in Figure 11 and Appendix A. In the worst case (i.e., a 180-s GNSS outage), the 2D position and height uncertainties in the forward KF processing are almost 23 and 21 times worse than in smooth processing, respectively. The 2D position and height uncertainties plots in smooth processing are like a bell-shaped diagram that increases with GNSS signal outage duration, but in forward KF processing, they are almost exponentially increased, and the positional error fixes after GNSS signal outage periods. For instance, the positional uncertainty using smooth processing after 180 s of GNSS outage goes back to the same uncertainty as before the GNSS signal outage (the same uncertainty diagram behavior can be seen for other GNSS outage durations too).

This is also true for 2D position and height uncertainties in the forward KF processing. Adding or removing the GLONASS data in the smooth and forward KF methods does not significantly change 2D position and height uncertainties. However, five GLONASS and nine GPS satellites were available during the GNSS signal outage duration. In other words, considering a period of 180 s, only the GLONASS outage provides the same results as without the signal outage (GPS + GLONASS).

The 2D position uncertainty considering a 180 s GPS outage increases up to 25 cm in smooth processing mode, but in forward KF processing mode, the 2D position uncertainty increases till 45 s, and after that, it follows approximately a constant (≈0.5 m) value (see Figure 10 and Appendix A). The height uncertainty for a 180 s GPS outage increases up to 43 cm in smooth processing mode (Figure 11). However, the error increases employing the forward KF processing, and it becomes approximately stable (≈1 m error) after 70 s (Appendix A).

It is crucial to consider the impact of GNSS signal outages, as evidenced by comparing positional uncertainties achieved using smoothing and forward KF with different durations of GNSS signal outages with the absolute position accuracy (root mean square) of the POSAV system in smart-based post-processing, where horizontal and vertical accuracies are 0.02 m and 0.05 m, respectively [53]. This comparison highlights the importance of accounting for GNSS signal outages in calculating positional EOPs to ensure accurate and reliable results.

#### 3.2.2. Smooth and Forward KF Processing Comparison Using Orientation Uncertainties

The impact of the GNSS signal outage on smooth and forward KF in the roll, pitch, and heading can be seen in Figure 12, Figure 13, Figure 14 and Appendix A, respectively. The roll, pitch, and heading uncertainties using forward KF processing in the worst case (180 s GNSS signal outage) are almost 2.5 times worse than smooth processing. Including or excluding the GLONASS data on smooth and forward KF processing does not significantly affect the roll, pitch, and heading uncertainties (i.e., a 180-s GLONASS signal outage is the same as a GNSS signal outage).

In addition, there is no significant difference between the best and worst cases (i.e., without a 180-s GNSS outage) using smooth processing. The corresponding errors obtained are 0.229 (0.00381 degree), 0.235 (0.00391 degree), and 0.595 (0.00991 degree) arc minutes, respectively, for roll, pitch, and heading, assuming no GNSS outage. Considering the worst-case scenario (i.e., a 180-s GNSS outage), the errors are 0.241 (0.00401 degree), 0.249 (0.00415 degree), and 0.605 (0.01 degree) arc minutes, respectively, for roll, pitch, and heading. A comparative analysis was conducted to evaluate the performance of the smoothing and forward Kalman filter techniques under varying durations of GNSS signal outages, along with the absolute orientational accuracy (root mean square) of a Position and Orientation System for Air Vehicles (POSAV) implemented in SmartBase post-processing [53]. The analysis revealed that the orientation uncertainties produced by both the best-case scenario (without GNSS signal outage) and the worst-case scenario (with 180 s of GNSS signal outage), as well as all other GNSS signal outage durations, fell within the range of the POSAV system’s orientational absolute accuracy.

However, upon comparing the orientation uncertainties generated by the forward KF technique during the worst-case scenario (180 s GNSS signal outage), namely 0.607 arcmin (0.0101 degrees), 0.614 arcmin (0.0102 degrees), and 1.260 arcmin (0.026 degrees) for roll, pitch, and heading, respectively, with the POSAV system’s orientational absolute accuracy. It was observed that the corresponding errors produced by the forward KF technique were almost twice as large as the POSAV system’s orientational absolute accuracy.

GNSS signal outages have a more pronounced effect on positional uncertainties compared to orientation uncertainties during smooth processing, primarily due to their impact on position determination. Orientation parameters such as roll, pitch, and heading are less affected, as they rely on other sensor data like accelerometers and gyroscopes. This resilience can be attributed to measurement availability and the use of sensor fusion techniques in attitude estimation. Attitude estimation depends on a combination of sensor measurements, which are generally more accessible and less susceptible to outages than GNSS measurements. Sensor fusion techniques, such as complementary filters, Kalman filters, or algorithms like Madgwick or Mahony filters, leverage redundant or complementary information from different sensors to mitigate the impact of intermittent GNSS signal outages, resulting in a more robust attitude estimate [9,10]. However, variations may exist in how different orientation parameters are affected by GNSS signal outages. For example, the graphs for roll and pitch uncertainties in smooth and forward KF response to a 180-s GPS signal outage may not follow the same pattern as other GNSS signal outage diagrams (Figure 12, Figure 13 and Appendix A).

In terms of the impact of adding GLONASS data on heading uncertainties, the information suggests that this may not improve heading uncertainties as much as it does for roll and pitch. The obtained heading uncertainties using a 180 s GPS signal outage and a 180 s GNSS (GPS + GLONASS) are close in both smooth and forward KF processing modes (Figure 14 and Appendix A). This suggests that adding GLONASS data may not significantly improve the accuracy of heading calculations compared to using GPS data alone. However, it is possible that adding GLONASS data may have other benefits, such as improving overall position accuracy or reducing the impact of other types of sensor data deficiencies.

In comparing our research with existing work in the field, Elamin et al. (2022) [54] demonstrated that their integrated GNSS/INS/LiDAR system achieved greater improvements in position. However, our study highlights the effectiveness of our smoothing algorithm in mitigating the impact of GNSS signal outages on both position and orientation uncertainties. Elbahnasawy et al. (2018) [31] present a low-cost UAV-based mapping system that integrates GNSS/INS with a spinning multi-beam LiDAR and a digital camera. The strategy effectively refines the trajectory during GNSS outages, resulting in significant improvements of 50% in planimetric accuracy, 25% in vertical accuracy, and 25% in heading accuracy. In comparison, while both studies address the impact of GNSS signal outages on positioning and orientation accuracy, they differ in their specific methods and approaches, providing valuable insights into the field of GNSS-based positioning and orientation. Also, Niu et al. (2014) [47] present Cinertial, a GNSS/INS data processing software, discussing its algorithm design, which encompasses INS mechanization, Kalman filtering, and backward smoothing. Their findings demonstrate that Cinertial achieves comparable accuracy (close to the software that we used in this study: POSPac MMS) to established commercial software solutions. A GNSS signal outage investigated by Zhou et al. (2015) [55] affects GNSS/INS integration. Without GNSS, the algorithm loses crucial spatial diversity information from multiple antenna elements, impacting accurate attitude estimation. During an outage, attitude RMSE may significantly increase until GNSS signals are restored or alternative sources are used. Our study relies more on inertial sensors, while Zhou’s study heavily depends on GNSS signals for attitude estimation.

### 3.3. Kalman Filter

A functional relationship (measurement model) between measurements and unknown parameters (system dynamic vector) is established for estimating unknown parameters, e.g., using the KF method. In cases where there are not enough measurements to estimate the unknown parameters, the dynamic model can also be used. Equations (5) and (6) are measurement and dynamic models [51].
(5)$$\underline{Z}(t)=H(t)\times \underline{x}(t)+\underline{\eta }(t)$$
(6)$$\underline{\dot{x}}(t)=F(t)\cdot \underline{x}(t)+G(t)\cdot \underline{w}(t)$$
where $\underline{Z}(t)$, $\underline{x}(t)$, and $\underline{\eta }(t)$ are the measurement vector, the state vector, and the measurement noise at the time t, respectively. H(t) is the geometry of the measurement with respect to the state vector (design matrix). $\underline{\dot{x}}(t)$ is the time derivation of $\underline{x}(t)$. F(t), G(t), and $\underline{w}\left(t\right)$ are system dynamic matrix, shape matrix (shaping the white noise in input to match the actual characteristic of the system), and process driving noise at time t, respectively. KF process in a discrete form can be rewritten as follows:(7)$${\underline{Z}}_{k+1}=H_{k+1}\times {\underline{x}}_{k+1}+{\underline{\eta }}_{k+1}$$
(8)$${\underline{x}}_{k+1}=\Phi_{k+1,k}\times {\underline{x}}_{k}+{\underline{w}}_{k+1}$$

$\Phi_{k+1,k}$ is the transition matrix in the dynamic model from epoch kth to (k+1)th, which is obtained from the continuous system dynamic matrix F in Equation (6).

KF is a recursive algorithm that aims to compute an optimal state vector with minimum variance by utilizing a sequence of prediction and update steps [51]. Figure 15 demonstrates the basic concept of KF, including the prediction and update steps.

The prediction equations (prediction step) estimate the state vector and the associated covariance matrix from the present to the next epoch. Superscripts “−” and “+” indicated a priori (before the update) and posteriori (after the update), respectively. $P_{k}$ is the covariance matrix of the state vector at epoch k and $Q_{k}$ is the covariance matrix of process noise.

Correction equations (update step) correct the state vector and associated covariance matrix based on the measurement model. $K_{k}$ is the Kalman gain matrix at epoch k and is calculated by minimizing state vector variance. ${\underline{v}}_{\;k}$ is the difference between the measurement vector and the predicted measurement vector called the innovation vector. The Kalman gain is a weighting factor utilized to show how new information incorporated in the innovation vector affects the final state vector estimation [56].

## 4. Conclusions

In conclusion, this study investigated the impact of a simulated GNSS signal outage on exterior orientation parameters (EOPs) solutions using tightly coupled GNSS/IMU integration with a forward Kalman filter and smoothing algorithm. Our results showed that the smoothing algorithm consistently outperforms the forward Kalman filter in terms of position and orientation accuracy, reducing uncertainties by up to 43% and 60%, respectively. Moreover, we found that using a network-based PPK solution provides more robust EOP calculations compared to a single station PPK solution.

Furthermore, we found that the GLONASS signal outage did not significantly affect EOP’s calculations, indicating that even without GLONASS satellites, the solution was robust and accurate enough due to the quantity and distribution of GPS satellites. This finding may have practical implications for GNSS users who operate in areas where GLONASS signal availability is limited or nonexistent.

Future studies could focus on improving the accuracy of EOP solutions under challenging GNSS signal conditions using other integration methods, such as loosely coupled integration and multi-sensor fusion, and exploring the use of other satellite systems such as Galileo and BeiDou. Overall, the findings of this study can be applied in various fields, such as airborne mobile mapping (manned or unmanned aerial photogrammetry), surveying, and geodesy, where accurate and robust positioning and orientation solutions (EOPs) are crucial.

## Figures and Tables

**Figure 1 sensors-23-06361-f001:**
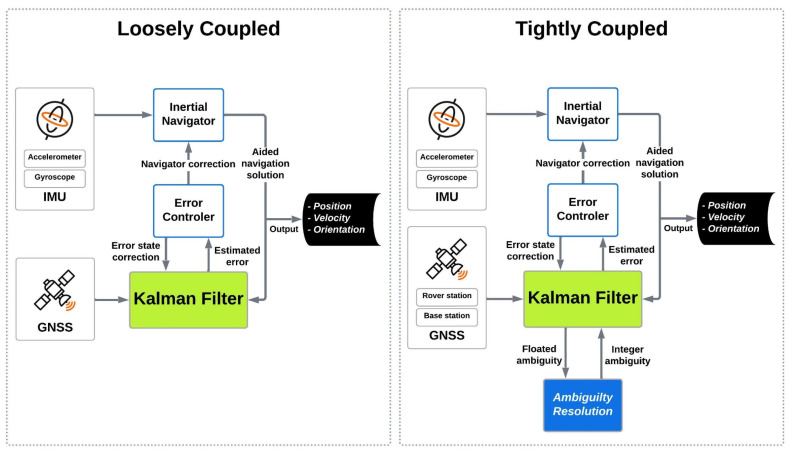
Tightly and Loosely coupled scheme. Note: Primary GNSS receiver is a GNSS receiver mounted on the aircraft (rover antenna). Base GNSS receivers are the receivers mounted on the ground to establish a network of GNSS stations. In a tightly coupled scheme, each of the primary and base GNSS data adds independently to the Kalman filter.

**Figure 2 sensors-23-06361-f002:**
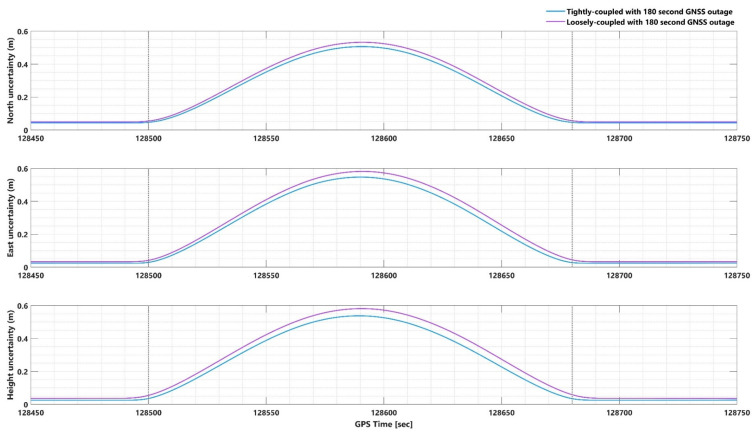
Position uncertainties in tightly- and loosely-coupled network-based PPK with smoothing algorithm for 180 s GNSS signal outage.

**Figure 3 sensors-23-06361-f003:**
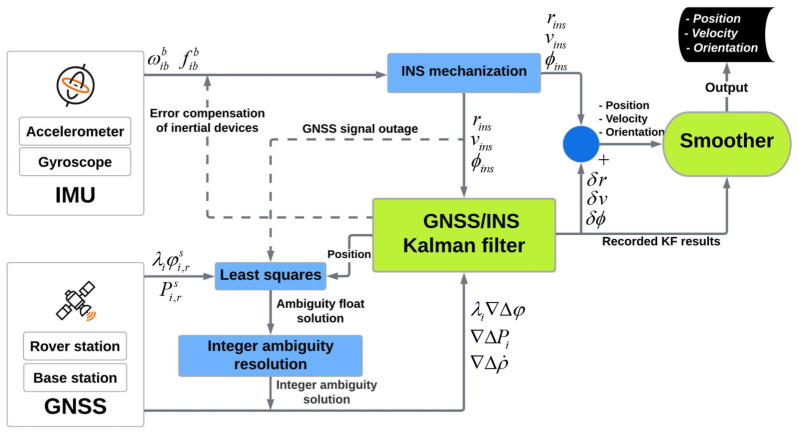
TC GNSS/INS integration data processing strategy, modified after [33]. Note: The subscripts r and S represent the GNSS receiver and satellite number, respectively. $\lambda_{i}\varphi_{i,r}^{s}$ and $P_{i,r}^{s}$ are observations of the raw carrier phase and pseudo-range, respectively. $f_{ib}^{b}$ is the accelerometer observation of specific force and $\omega_{ib}^{b}$ is the gyroscope observation of angular rate (*i* denotes inertial-frame and *b* is IMU body-frame [34]). The position, velocity, and orientation that are estimated by INS are shown with $r_{ins}$, $v_{ins}$, and $\phi_{ins}$, respectively. δr, δv, and δϕ are their corresponding correction. $\lambda_{i}\nabla \Delta \varphi$, $\nabla \Delta P_{i}$, and $\nabla \Delta {\dot{\rho }}_{i}$ are the double difference observation of carrier phase, pseudorange, and pseudorange rate, respectively.

**Figure 4 sensors-23-06361-f004:**
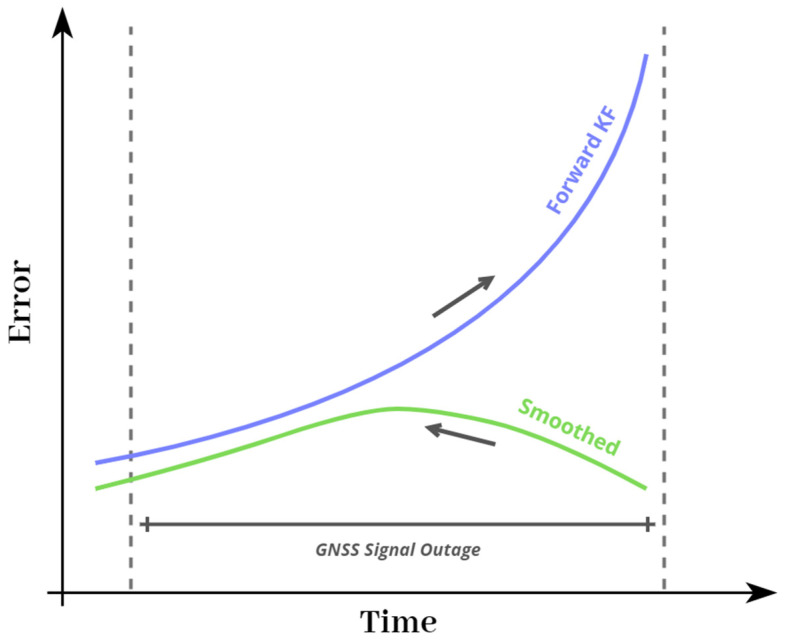
A schematic illustration of forward KF and smoothing error curve behavior of EOPs during GNSS signal outage.

**Figure 5 sensors-23-06361-f005:**
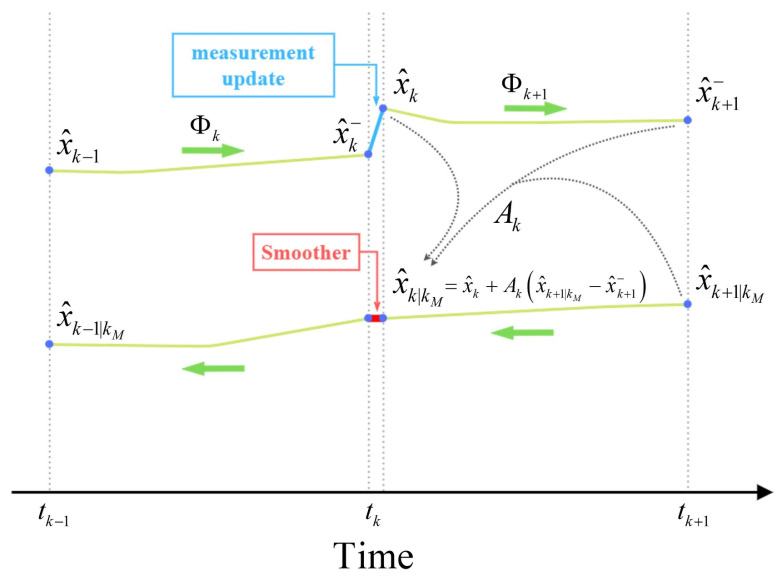
Forward and backward run of the smoother.

**Figure 6 sensors-23-06361-f006:**
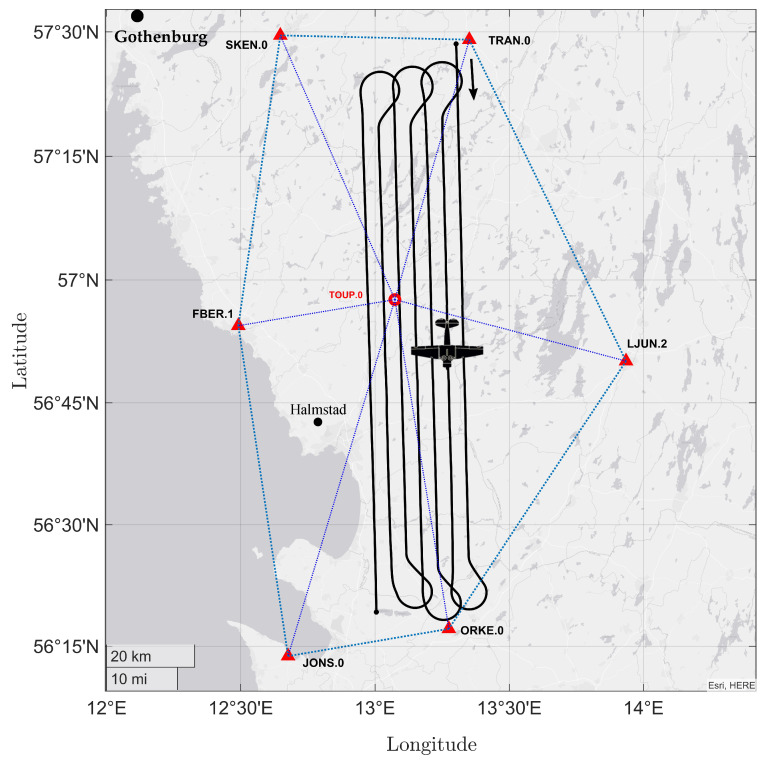
The trajectory of the rover (aircraft) and position of the SWEPOS stations in the study area.

**Figure 7 sensors-23-06361-f007:**
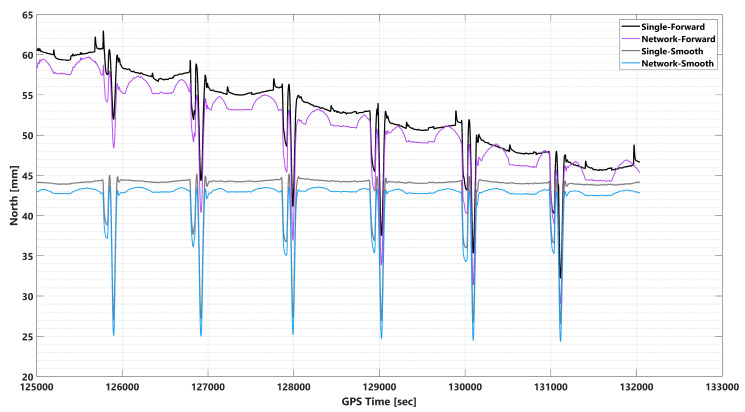
North component uncertainties in single-based PPK with forward KF, network-based PPK with forward KF, single-based PPK with smooth processing, and network-based PPK smooth processing modes.

**Figure 8 sensors-23-06361-f008:**
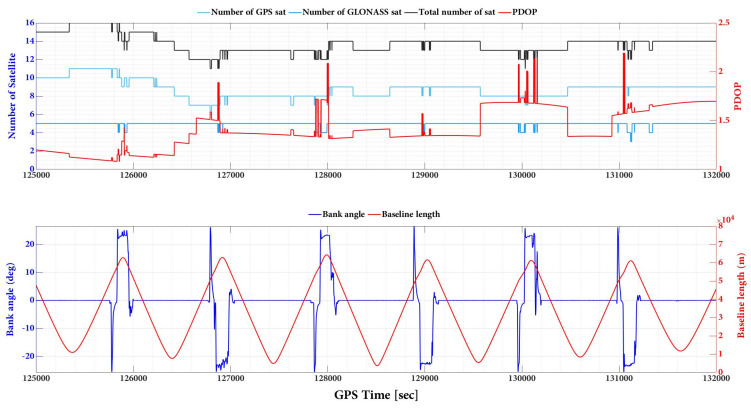
The number of satellites, PDOP, bank angle, and baseline length plots during the flight mission. Note: The baseline length is the distance between the rover (aircraft) and the assigned GNSS reference station on the ground (TOUP.0).

**Figure 9 sensors-23-06361-f009:**
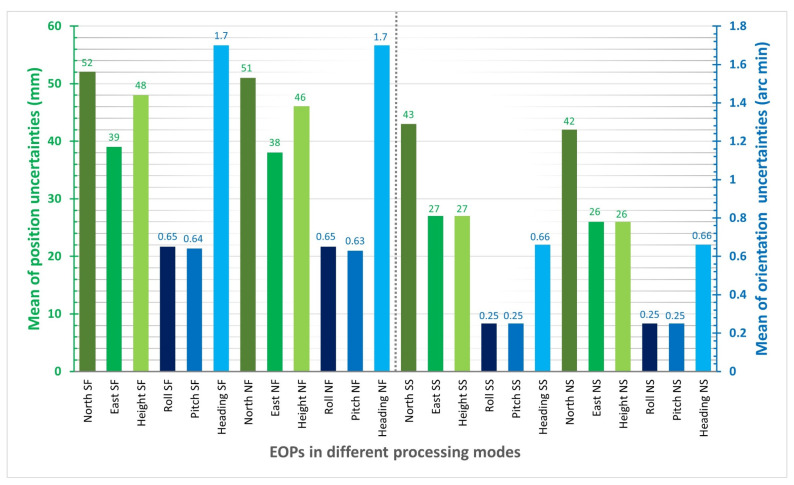
Mean of EOPs uncertainties in different processing modes, i.e., single-based PPK with forward KF (SF), network-based PPK with forward KF (NF), single-based PPK with smooth processing (SS), and network-based with smooth processing (NS).

**Figure 10 sensors-23-06361-f010:**
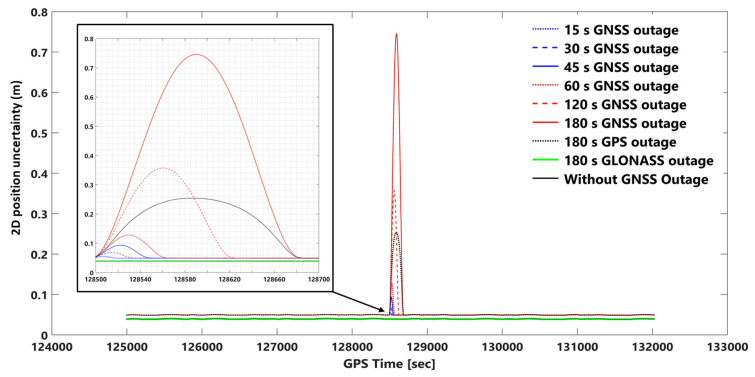
2D position (north and east) uncertainties using network-based PPK with smooth processing and assuming different GNSS (GPS + GLONASS) signal outages. Note: for better illustration: The inset figure shows the focused zoom of the main plot during the GNSS signal outage. We set a −0.01 m shift for 180 s GLONASS outage and without GNSS outage plots for both inset and whole mission plots to make a better illustration.

**Figure 11 sensors-23-06361-f011:**
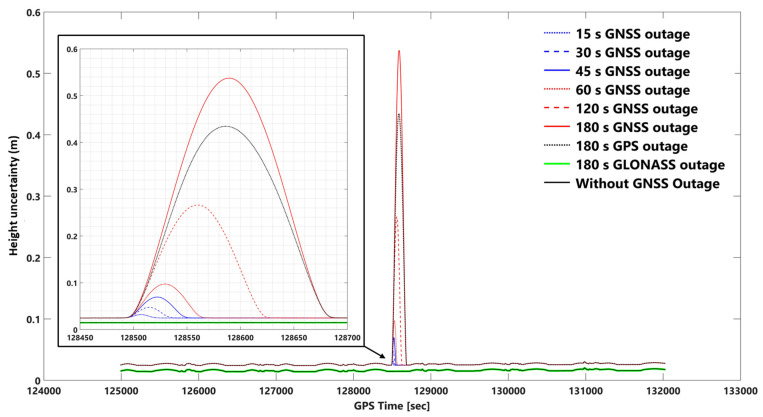
Height uncertainties using network-based PPK with smooth processing and assuming different GNSS (GPS + GLONASS) signal outages. Note: for better illustration: The inset figure shows the focused zoom of the main plot during the GNSS signal outage. We set a −0.01 m shift for 180 s GLONASS outage and without GNSS outage plots for both inset and whole mission plots to make a better illustration.

**Figure 12 sensors-23-06361-f012:**
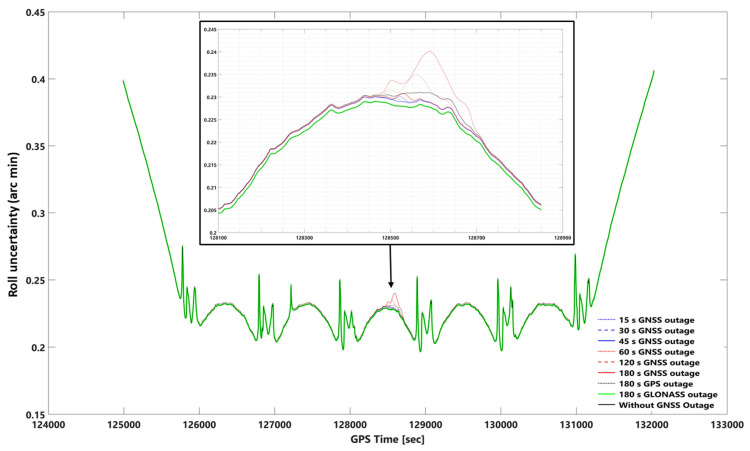
Uncertainties of the roll angle using network-based PPK with smooth processing and assuming different GNSS (GPS + GLONASS) signal outages. Note: for better illustration: The inset figure shows the focused zoom of the main plot during the GNSS signal outage. We set −0.001 arc minute shift for 180 s GLONASS outage and without GNSS outage plots for both inset and whole mission plots to make a better illustration.

**Figure 13 sensors-23-06361-f013:**
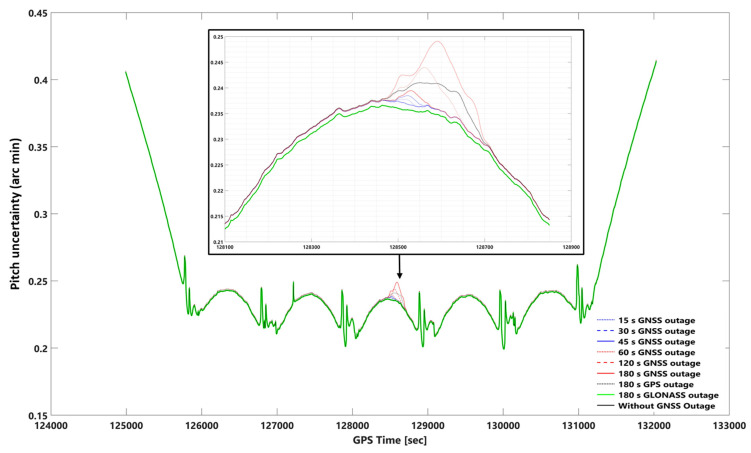
Uncertainties of the pitch using network-based PPK with smooth processing and assuming different GNSS (GPS + GLONASS) signal outages. Note: for better illustration: The inset figure shows the focused zoom of the main plot during the GNSS signal outage. We set −0.001 arc minute shift for 180 s GLONASS outage and without GNSS outage plots for both inset and whole mission plots.

**Figure 14 sensors-23-06361-f014:**
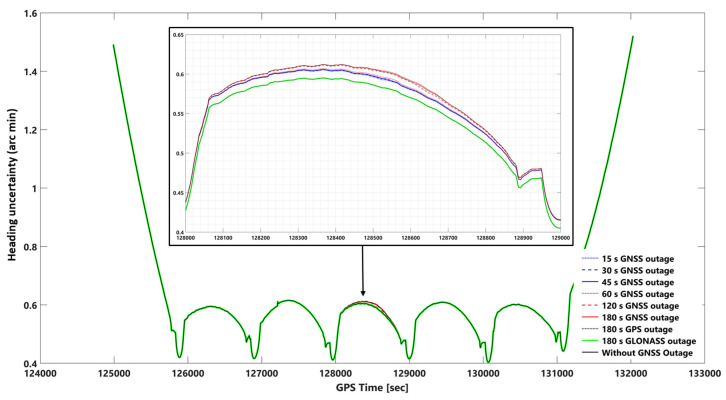
Uncertainties of heading using network-based PPK with smooth processing and assuming different GNSS (GPS + GLONASS) signal outages. Note: for better illustration: The inset figure shows the focused zoom of the main plot during the GNSS signal outage. We set −0.01 arc minute shift for 180 s GLONASS outage and without GNSS outage plots for both inset and whole mission plots to make a better illustration.

**Figure 15 sensors-23-06361-f015:**
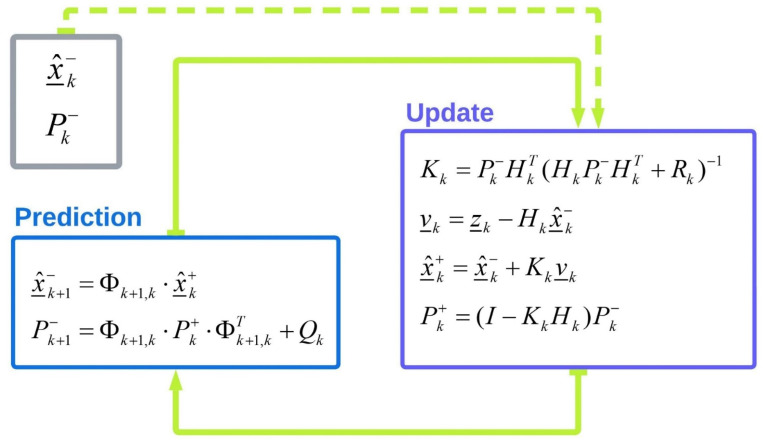
Schematic illustration of KF algorithm’s steps.

## Data Availability

Some or all data, models, or codes that support the findings of this study are available from the corresponding author upon reasonable request.

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
