# Peer review of "Numerical Analysis of GNSS Signal Outage Effect on EOPs Solutions Using Tightly Coupled GNSS/IMU Integration: A Simulated Case Study in Sweden"

_sensors, 2023, doi:10.3390/s23146361_

Round 1

Reviewer 2 Report

This paper analyzes the performance of tightly coupled GNSS/INS with an emphasize on the exterior orientation parameters. The contribution of paper this paper is very small indeed. The experiment result seems to be sound. The following suggestion should be considered.

1. The contribution of the work should be compared with existing literatures, especially on the GNSS outage effect on exterior orientation parameters.

2.  The filtering model of tightly coupled GNSS/INS should be given, and the employed filter and smoother should also be reviewed briefly.

3. Some basic knowledge on smoother and integration structure of GNSS/INS should be deleted.

4. I think it is better to remove the “180s GLONASS outage” legend from Figure 10-14 (it confuse me at the first looking), and just points out it in the result description.

5.  The reviewer can’t find Figures S1-S10.

6.  More discussion on the experiment result should be added, i.e., why the outages affect the attitude less compared with position from the filter or smoother viewpoint.

Round 2

Reviewer 1 Report

The author has made the revision.

Author Response

We extend our sincerest gratitude for your valuable and constructive commentary provided during the preceding reviewing cycle. Regrettably, we do not possess a response to offer, as the reviewer expressed contentment with our initial round of responses.

Reviewer 2 Report

The authors does not addressed my comments. The follow suggestions may be of some help. The main contribution of this paper is still unclear. The specific filtering model of tightly coupled GNSS/INS is not given and the description of smoothed algorithm is rather basic. Why the authors compare loosely coupled with tightly coupled structure, as there is no controversy on that the later performs better regarding robustness.  The result and contribution discussion should focus on the experiment result of this paper. The structure of this paper is not well arranged.
